# SIEVE: General Purpose Data Filtering System Matching GPT-4o Accuracy at 1% the Cost

## Abstract

Creating specialized large language models requires vast amounts of clean, special purpose data for training and fine-tuning. With only a handful of existing large-scale, domain-specific datasets, creation of new datasets is required in most applications. This requires the development of new application-specific filtering of web-scale data. Filtering with a high-performance, general-purpose LLM such as GPT-4o can be highly effective, but this is extremely expensive at web-scale. This paper proposes SIEVE, a lightweight alternative that matches GPT-4o accuracy at a fraction of the cost. SIEVE can perform up to 500 filtering operations for the cost of one GPT-4o filtering call. The key to SIEVE is a seamless integration of GPT-4o and lightweight T5 models, using active learning to fine-tune T5 in the background with a small number of calls to GPT-4o. Once trained, it performs as well as GPT-4o at a tiny fraction of the cost. We experimentally validate SIEVE on the OpenWebText dataset, using five highly customized filter tasks targeting high quality and domain-specific content. Our results demonstrate the effectiveness and efficiency of our method in curating large, high-quality datasets for language model training at a substantially lower cost (1%) than existing techniques. To further validate SIEVE, experiments show that SIEVE and GPT-4o achieve similar accuracy, with human evaluators preferring SIEVE's filtering results to those of GPT-4o.

## 1 Introduction

Large Language Models (LLMs) have revolutionized natural language processing, demonstrating remarkable capabilities across a wide range of tasks. As the field progresses, there is growing interest in developing specialized LLMs tailored to specific domains or applications (Lee et al., 2023; Gupta et al., 2024; Li et al., 2024). A critical component in this process is the curation of high-quality, domain-specific datasets for training and fine-tuning these models. However, the task of filtering vast amounts of web-scale data to create such datasets presents significant challenges in terms of cost, time, and effectiveness. Existing approaches to data filtering and curation primarily rely on two strategies: (1) sourcing data from specific, trusted sources, and (2) employing pre-existing quality and toxicity detection models to filter harmful content. For instance, medical LLMs often utilize datasets derived from PubMed to ensure domain specificity. While these data filtering methods have their merits, they suffer from notable limitations in terms of both flexibility and comprehensiveness. Many domains lack comprehensive, exclusive sources of high-quality text data, and pre-trained detection models typically focus on general quality metrics and toxicity, limiting their applicability for domain-specific queries or customized quality rubrics. Furthermore, relying solely on established data sources ignores the vast amount of potentially valuable information available on the broader internet, leading to datasets that may be limited in scope and diversity. These constraints can significantly hinder the development of specialized language models that require rich, domain-specific training data.

Recent advancements in general-purpose language models, such as GPT-4o, offer a potential solution to these challenges. These models can act as effective filtering mechanisms when provided with appropriate filtering prompts. To illustrate as an example, one could employ GPT-4o to iterate over all text snippets of the internet to determine whether each piece pertains to the 2024 presidential election. In this context, the machine learning model serves as a binary classification mechanism, evaluating one snippet at a time. However, the computational cost of applying models like GPT-4o to web-scale datasets is prohibitively expensive for most organizations. In this paper, we present a novel system SIEVE that addresses these limitations, providing versatile and high-quality data

Figure 1: **System Overview.** From user's perspective, SIEVE acts as if applying GPT-4o with the filtering prompt to all text snippets in a web-scale dataset. The output from the SIEVE system is the set of all text snippets that receive a 'pass'. To reduce the prohibitively high cost of applying GPT-4o on every snippet, SIEVE utilizes active learning to distill lightweight filtering models based on pretrained T5 encoders, effectively reducing the overall cost to less than 1%.

filtering at a fraction of the cost of using high performance, general-purpose models directly. Shown in Figure 2, our method is based on a lightweight T5 model in combination with a small number of calls to GPT-4o, effectively reducing the cost per filtering to one call to T5 instead of GPT-4o. This is accomplished by training and distilling a lightweight T5 model in the background that learns to mimic the filtering decisions of GPT-4o and eventually handles all/most of the filtering. This background job is optimized via a novel active learning algorithm that further reduces the need to call GPT-4o.

Our active learning algorithm operates in a stream-based setting, sequentially and adaptively selecting the most informative text snippets for evaluation by GPT-4o. Inspired by previous work in deep active learning (Zhang et al., 2022a; Nuggehalli et al., 2023), we propose a novel stream-based algorithm designed to tackle scenarios where the data distribution is imbalanced. As all of the filtering decisions are highly imbalanced (see Table 3), one of our main contributions is proposing the algorithm to tackle this imbalance issue. Our algorithm is designed to efficiently label a set of more balanced and uncertain examples. As demonstrated in our experiments, SIEVE achieves GPT-4o quality filtering performance at less than 1% of the cost. In addition, compared to random sampling methods, our active learning algorithm reduces the number of queries to GPT-4o by more than 5x (see Section 5). In Section 4, we provide theoretical analysis of our algorithm, establishing formal proofs of balancedness bounds for active learning in scenarios with imbalanced underlying data distributions. To the best of our knowledge, this represents the first attempt to provide rigorous theoretical guarantees in this context.

The implications of SIEVE are far-reaching, democratizing access to clean, task-specific data and facilitating the development of specialized language models across various domains. By dramatically reducing the cost and complexity of data filtering, SIEVE opens new avenues for researchers and organizations to create tailored LLMs for specific applications, industries, or scientific disciplines.

In summary, the core contributions of this work are:

- A novel, cost-effective system, SIEVE, for high-quality data filtering that achieves GPT-4o level performance at less than 1% of the cost.
- A stream-based active learning algorithm for imbalanced datasets that efficiently selects informative snippets, reducing the number of queries to GPT-4o more than 5x compared to random sampling.
- Theoretical analysis and formal proofs of balancedness bounds for our active learning algorithm in scenarios with imbalanced data distributions.
- Experimental validation of SIEVE on the OpenWebText dataset using five highly specific filters, demonstrating its effectiveness and versatility.

## 2 A GENERAL PURPOSE DATA FILTERING SYSTEM

In this section we provide a detailed description of the SIEVE system. A visualization of our system can be found in Figure 1.

**A Bird's-Eye View.** When viewed as a black box, SIEVE processes a web-scale dataset of $N$ text snippets along with a filtering prompt that specifies the criteria for passing or failing each snippet. This prompt specifies the criteria for passing or failing each snippet, similar to how one would instruct any high-performance, general-purpose LLM, such as GPT-4o. From this perspective, SIEVE efficiently categorizes each snippet as 'pass' or 'fail' based on the provided prompt, with filtering quality comparable to that of GPT-4o.

A straightforward approach would be to apply GPT-4o with the filtering prompt to each text snippet. However, this becomes extremely costly for web-scale datasets containing billions or trillions of tokens. For example, filtering the OpenWebText dataset (Gokaslan & Cohen, 2019) used in this study, which contains 9 billion tokens, would cost approximately $67,000 using GPT-4o directly. For even larger datasets like the PILE (Gao et al., 2020), this cost would increase by at least 1000 times, making it prohibitively expensive. To overcome this challenge, we utilize an active distillation framework as detailed in the following section.

## 2.1 An Active Distillation Framework

In this section, we describe the inner workings of SIEVE, which employs a lightweight binary classification model trained on GPT-4o's filtering decisions. By leveraging active learning techniques, we selectively gather GPT-4o decisions on a small, informative subset of text snippets from the web-scale dataset. This approach significantly reduces costs while maintaining filtering quality comparable to GPT-4o.

Active learning minimizes annotation costs from expensive sources by selecting the most informative subset of snippets to query for labels. The goal is to train a high-performance model $f$ with minimal annotation cost. In our framework, GPT-4o serves as the expensive annotation source, while we fine-tune a pretrained T5 encoder model for binary classification on the collected data and annotations to achieve high performance. Active learning strategies collect annotations incrementally, retraining the lightweight T5 model $f$ after labeling every $B$ new snippets. While most deep active learning literature focuses on the pool-based setting (Ash et al., 2019; Nuggehalli et al., 2023; Fairstein et al., 2024; Lesci & Vlachos, 2024), these algorithms require forward inference of $f$ on the entire dataset at every iteration, incurring high computational costs for large-scale datasets (see Section 5.3 for details). To mitigate this, we apply active learning in the streaming setting, an area well-studied classically but with limited research for neural networks. We specifically designed our algorithm to tackle the imbalance in filtering decisions generated by GPT-4o (see Table 3), aiming to query GPT-4o on a more balanced and informative set of text snippets.

Formally, we assume access to a stream of i.i.d. drawn snippets $x_1, x_2, ..., x_N$, all following the same underlying data distribution $\mathbb{P}_X$. We let $S = x_1, ..., x_N$ denote the stream. In practice, we construct the stream by randomly shuffling all snippets in the OpenWebText dataset (Gokaslan & Cohen, 2019). At time $i$, the active learning algorithm observes $x_i \sim \mathbb{P}_X$ and decides whether to query GPT-4o for its annotation based on the snippet's informativeness to model $f$. If queried, we obtain the corresponding filtering decision from GPT-4o, which we denote by $y_{GPT}(x_i) \in \{0, 1\}$. Here, the randomness comes from the nondeterministic nature of GPT-4o's response. After every $B$ new annotations, we fine-tune the T5-decoder model from its pretrained checkpoint to obtain an updated model $f$. The distillation process terminates after a total annotation budget of $T$ snippets. The final model $f$ fine-tuned on queried snippets is then applied to filter the entire web-scale dataset.

## 3 Active Learning Algorithm for Distilling GPT-4o

In this section, we present a novel stream-based active learning algorithm for class-imbalanced scenarios. We use a modified version of the uncertainty sampling strategy (Lewis & Gale, 1994; Tong & Koller, 2001; Balcan et al., 2006; Settles, 2009; Kremer et al., 2014). In our setting, uncertainty sampling labels snippets $x_i$ that have predictive sigmoid score around 0.5, i.e. $f(x_i) \approx 0.5$, as these are believed to be the most informative data to be labeled. However, our binary classification tasks of data filtering is naturally imbalanced as shown in Table 3. Previous active learning work by Zhang et al. (2022a) and Nuggehalli et al. (2023) observed that the threshold of 0.5 is generally biased towards the majority class under imbalanced scenarios, which means that most of the snippets selected by uncertainty sampling will be in the majority class. This translates into poor performance

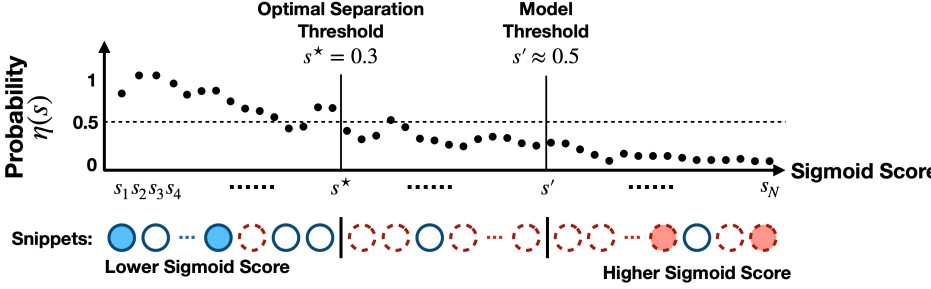

Figure 2: Demonstration of the TRM threshold. Snippets (shown on the bottom) are first ordered based on their predictive sigmoid scores. GPT-4o class labels $0$ and $1$ are represented by the solid or dashed borders. Queried snippets are shaded. Under imbalanced scenarios, sigmoid score of $0.5$ generally will not provide a good indication of where to sample, and will likely result in labeling much more snippets in the majority class. The probability $\mu(s)$ denotes the likelihood of a snippet with sigmoid score $s$ belonging to class $0$. The TRM threshold is defined to best separate the two classes of snippets.

---

**Algorithm 1** Stream-Based Class-Balancing Active Learning

---

**Input:** Data Stream $S$, labeling function $y_{GPT}$ based on GPT-4o and specified filtering prompt, batch size $B$, total budget $T$ and confidence level $\delta$.

**Initialize:** Query $x_1, ..., x_B \stackrel{iid}{\sim} \mathbb{P}_X$ to form the initial labeled set $L = \{x_i, y_i\}_{i=1}^B$.

**for** $r = 1, ..., \frac{T}{B}$ **do**
    Fine-tune pretrained T5 encoder model on the latest labeled set $L$ to form $f : S \to [0, 1]$.
    Initialize confidence set $\underline{\mu}, \bar{\mu} \leftarrow 0, 1$, counter $t \leftarrow 0$.
    Let $j$ index the head of the stream $S$, $x_j$.
    **while** $|L| < (s + 1)B$ **do**           // **Find optimal separation threshold for $f$.**
        Receive the next snippet in the stream $S$, $x_{j+t} \sim \mathbb{P}_X$.
        **if** $f(x_{j+t}) \in [\underline{\mu}, \bar{\mu}]$ **then**
            Query GPT-4o for label $y_{j+t}$ by $y_{GPT}$, and insert to set $L \leftarrow L \cup \{x_{j+t}, y_{j+t}\}$.
        **end if**
        **if** $t \in 2^{\mathbb{N}^+}$ **then**                  // **Update confidence interval.**
            Store previously computed sigmoid scores $F_t \leftarrow \{0, f(x_j), ..., f(x_{j+t})\}$.
            Compute the empirical TRM threshold as $\widehat{s}_t \leftarrow \min_{s \in F_t} \widehat{\mathcal{L}}_t(s)$, where
            $\widehat{\mathcal{L}}_t(s) := \frac{1}{t+1}(\sum_{i \in [j,j+t]:f(x_i) \leq s} \mathbf{1}\{y_i \neq 0\} + \sum_{i \in [j,j+t]:f(x_i) > s} \mathbf{1}\{y_i \neq 1\})$.
            Update $\underline{\mu}, \bar{\mu}$ be the smallest and largest thresholds $s \in F_t$ such that

$$\widehat{\mathcal{L}}_t(s) - \widehat{\mathcal{L}}_t(\widehat{s}_t) \leq \beta_{t+1}\sqrt{(|\{s' \in F_t : \min(s, \widehat{s}_t) \leq s' \leq \max(s, \widehat{s}_t)\}| - 1)/t} + \beta_{t+1}^2/2$$

            where $\beta_{t+1} = \sqrt{2\log(2\log_2(t+1)^2 N^2/\delta)/(t+1))}$ is chosen so that the TRM threshold lies in the updated confidence threshold with probability of at least $1 - \delta/\log_2(N)$.
        **end if**
        Update counter $t \leftarrow t + 1$.              // **Loop over snippets in S.**
    **end while**
**end for**
**Return:** T5 encoder model $f$ finetuned on $L$. $f$ is then used for filtering on the entire dataset.

---

in detecting snippets from the target minority class. To combat this, our approach instead aims to find a threshold near where the majority and minority classes are equiprobable, which we will refer to as an *True Risk Minimizer (TRM) threshold*. Selecting and labeling snippets around the TRM threshold yields a labeled dataset that is more balanced and includes uncertain snippets. In a nutshell, our algorithm (detailed in Algorithm 1) alternates between

1. Spending $B$ budget in finding and labeling close to the TRM threshold;
2. Fine-tuning a new T5 encoder model $f$ on all labeled snippets thus far.

More concretely as shown in Figure 2, when ordering snippets according to their sigmoid scores, labeling snippets around the threshold of $0.5$ can often result in selecting and labeling of most of snippets from a single (majority) class. Our proposed alternative, the TRM threshold, minimizes the expected number of class 1 snippets to its left and class 0 snippets to its right. Thus, the TRM threshold minimizes the expected number of misclassifications. Formally, let $\eta_j$ denote the probability of snippet $x_j$ belonging to class 0

$$\eta_j := \mathbb{P}\big(y_{\text{GPT}}(x_j) = 0\big)$$

where the possible randomness is with respect to GPT-4o's response as well as the distribution underlying the snippet $x_j \sim P_X$. The distribution $P_X$ is unknown, so $\{\eta_j\}$ are unknown as well. For model $f$ and a stream of i.i.d. text snippets $x_1, ..., x_N \sim \mathbb{P}_X$, the *True Risk Minimizer (TRM) threshold* is then the sigmoid score $s^\star \in \{0, f(x_1), \ldots, f(x_N)\}$, where

$$s^\star := \underset{s \in \{f(x_1),\ldots,f(x_N)\}}{\arg\min} \left( \sum_{j:f(x_j) \leq s} (1 - \eta_j) + \sum_{j:f(x_j) > s} \eta_j \right). \tag{1}$$

The optimization objective above can be viewed as the expected risk (probability of error) of a finite set of threshold classifiers with threshold locations at $\{0, f(x_1), \ldots, f(x_N)\}$. A threshold classifier at $s$ categorizes snippets into class 0 if their sigmoid score is less than or equal to $s$, and into class 1 if their score is greater than $s$. The TRM threshold can also be viewed as where the two classes best separate from each other. Snippets around this threshold are therefore truly uncertain to the neural network model. Moreover, in Section 4, we will provide a novel theoretical analysis demonstrating labeling around the TRM threshold also improves the balancedness of the queried snippets, alleviating the data imbalance issues when training the lightweight model $f$. Of course, the TRM threshold is unknown (because the probabilities $\{\eta_j\}$ are unknown), so we employ an efficient active learning procedure to identify a threshold close to it.

Our algorithm, shown in Algorithm 1, applies agnostic active learning techniques (Dasgupta et al., 2007; Jamieson & Jain, 2022; Katz-Samuels et al., 2021) to threshold classifiers. It focuses on identifying the TRM threshold. The algorithm initializes by querying the first $B$ snippets from the stream to create a labeled set $L$. Each iteration begins with fine-tuning classifier $f$ on $L$. To estimate the TRM threshold $s^\star$, we maintain a high-confidence interval $[\underline{\mu}, \bar{\mu}]$ while annotating stream snippets. Snippets with predictive sigmoid scores within this interval are queried using GPT-4o. We update the confidence interval on a geometric schedule, every $2^i$ snippets, using empirical estimates $\hat{s}_t$ of the optimal threshold. This interval, centered around the empirical estimate, ensures $s^\star \in [\underline{\mu}, \bar{\mu}]$ with at least $1 - \delta$ probability across all $t = 2^i$ for the current iteration. For constructing the confidence interval, we employ an empirical Bernstein bound, as introduced in Jamieson & Jain (2022). This approach differs from the original CAL algorithm (Dasgupta et al., 2007), which uses a uniform convergence bound requiring synchronous querying. Our method allows for parallel GPT-4o queries, significantly reducing computational time in practice.

## 4 THEORETICAL ANALYSIS

In this section, we first provide formal guarantee of the performance of our algorithm. We then proceed to analyze the balancedness of the snippets our algorithm queries, showing its improvement in collecting a more balanced set of snippets. Note that all of our analysis are conducted for any single iteration $s$ in Algorithm 1. Define the true risk function over all snippets $1, \ldots, N$ at threshold $s$ is denoted by

$$R(s) = \frac{1}{N} \left( \sum_{i \leq N: f(x_i) \leq s} (1 - \eta_i) + \sum_{i \leq N: f(x_i) > s} \eta_i \right).$$

Recall the goal is to find a threshold near $s^* = \arg\min_s R(s)$. This learning problem over the discrete set of threshold classifiers at thresholds $f(x_1), ..., f(x_N)$ can be exactly represented in the framework of Jamieson & Jain (2022), leading to Algorithm 1 and the following bound.

**Theorem 4.1** (Jamieson & Jain (2022)). *During iteration $r$ of Algorithm 1, given the classifier model $f$, with probability at least $1 - \delta$, both $R(\underline{\mu}) - R(s^\star)$ and $R(\bar{\mu}) - R(s^\star)$ are upper bounded by*

$$c_0 R(s^\star) + c_1 \beta_t \sqrt{R(s^\star)} + c_2 \beta_t^2. \tag{2}$$

*for all the confidences intervals $[\underline{\mu}, \bar{\mu}]$ updated at time $t \in 2^{N^+}$. Here, $c_0, c_1$ and $c_2$ are some universal constants.*

The theorem suggests that the gap in the true risk of the confidence intervals shrinks roughly on the scale of $\frac{1}{\sqrt{t}}$ over time since $\beta_t = \widetilde{O}(\frac{1}{\sqrt{t}})$. For large $t$, this gap goes to $0$.

**Definition 4.2** (Score Re-Ordering). Let $\pi$ denote the permutation of $\{1, \ldots, N\}$ such that $f(x_{\pi(1)}) \leq \cdots \leq f(x_{\pi(N)})$.

**Definition 4.3** (Discrete Smoothness). Let $L = \max_{j \in [N-1]} |\eta_{\pi(j)} - \eta_{\pi(j+1)}|$ denote the maximum change in probabilities $\eta_{(j)}$. This mirrors the Lipschitz smoothness in a discrete fashion. Note we always have $L \leq 1$.

Without loss of generality, assume class $0$ to be the minority class. The expected imbalance ratio is then $\frac{\sum_{j=1}^{N} \eta_j}{\sum_{j=1}^{N} 1 - \eta_j} < 1$. When $s^\star = 0$, it means even the lowest sigmoid score snippets are more likely to be in the majority class. In this case, a reasonable strategy is simply query the lowest sigmoid score snippets, which is exactly what our algorithm does.

**Theorem 4.4** (Balancedness of Labeled Snippets). *Assume class 0 is the minority class and $s^\star \neq 0$. Consider an interval of scores $[\underline{\mu}, \bar{\mu}]$ with $s^\star \in [\underline{\mu}, \bar{\mu}]$. Let the corresponding gaps in risk denoted by $\gamma_0 := R(\underline{\mu}) - R(s^\star) > 0$ and $\gamma_1 := R(\bar{\mu}) - R(s^\star) > 0$. When labeling snippets indexed within this interval uniformly at random, the imbalance ratio $\lambda(\underline{\mu}, \bar{\mu})$ between the minority class and the majority class must satisfy*

$$\lambda(\underline{\mu}, \bar{\mu}) = \frac{\sum_{j : f(x_j) \in [\underline{\mu}, \bar{\mu}]} \eta_j}{\sum_{j : f(x_j) \in [\underline{\mu}, \bar{\mu}]} 1 - \eta_j} \geq 1 - \min\left(\frac{N\bar{\gamma} + LN}{1.5 - 2L}, \sqrt{L} \cdot \frac{N\bar{\gamma} + LN + 1}{(1-L)\sqrt{N\underline{\gamma}}}\right)$$

*where $\underline{\gamma} := \min(\gamma_0, \gamma_1)$ and $\bar{\gamma} := \max(\gamma_0, \gamma_1)$, with $L, \underline{\gamma}, \bar{\gamma} < 1$. This implies,*

*(a) if $\bar{\gamma} \to 0$ and $L \to$, $\lambda(\underline{\mu}, \bar{\mu}) \geq 1 - \frac{N\bar{\gamma} + LN}{1.5 - 2L} \to 1$ (perfect balance);*

*(b) if $\underline{\gamma} \geq \frac{c}{N}$ for some constants $c > 0$, then $\lambda(\underline{\mu}, \bar{\mu}) \geq 1 - \sqrt{L} \cdot \frac{N\bar{\gamma} + LN + 1}{c(1-L)}$. When $L \to 0$, we again recovers $\lambda(\underline{\mu}, \bar{\mu}) \to 1$ (perfect balance).*

*Proof sketch.* Let $A = \sum_{j : f(x_j) \in [\underline{\mu}, \bar{\mu}]} \eta_j$ and $B = \sum_{j : f(x_j) \in [\underline{\mu}, \bar{\mu}]} 1 - 2\eta_j$, we can rewrite the imbalance ratio into $\lambda(\underline{\mu}, \bar{\mu}) = \frac{A}{A+B}$. Since $N(\gamma_1 - \gamma_0) = \sum_{j : f(x_j) \in (\underline{\mu}, \bar{\mu}]} 1 - 2\eta_j \geq B - LN$, we can lower bound the balancedness by $\frac{A}{A + N(\gamma_1 - \gamma_0) + LN}$. We therefore need to prove that $A$ is sufficiently large as compared to $N(\gamma_1 - \gamma_0) + LN$.

To prove this, we note that the $\eta$ values around the optimal separation threshold $s^\star$ are close to $.5$. Therefore, by the smoothness condition, when $L$ is small, $\eta_j \approx .5$ for all $f(x_j) \in [\underline{\mu}, \bar{\mu}]$, so $A$ roughly scales linearly in the number of elements within $[\underline{\mu}, \bar{\mu}]$. We can prove that there are at least $O(\sqrt{\frac{N\bar{\gamma}}{L}})$ elements in this confidence interval. $A$ therefore follows a similar scale, which is much greater than $N(\gamma_1 - \gamma_0) + LN$ when $L \to 0$. Under such case, we recover a balancedness bound of $1$, i.e. the annotated snippets are perfectly balanced. Our detailed proof is provided in Appendix B.

## 5 EXPERIMENTS

In this section, we present our experiments conducted for the five highly customized data filters in Table 1: politics, climate, AI, mainstream knowledge and text quality filters. These filters are applied to the OpenWebText dataset (Gokaslan & Cohen, 2019), which is divided into around 13.5M snippets of 1024 tokens. The first three filters identify text snippets related to a particular topic, with highly detailed specifications of filtering prompts to GPT-4o about the subdomain of topics that should be included.

| Filter | Imbalance Ratio $\lambda$ |
|---|---|
| Politics | 0.153 |
| Climate | 0.043 |
| AI | 0.026 |
| Mainstream | 0.208 |
| Quality | 0.457 |

Figure 3: Imbalance ratio of minority vs majority decisions, calculated based on 5000 randomly sampled snippets. $\lambda = $ #Minority / #Majority.

| Filter | Method | Bal. Accuracy (GPT-4o as GT) | Human Preference Over GPT-4o | #Queries to GPT-4o | Lightweight Model Cost | Total Cost |
|---|---|---|---|---|---|---|
| Politics | GPT-4o | **95.6%**[*] | — | 13.5M | $0 | $67,000 |
| | SIEVE (Ours) | **95.6%** | — | 60K | $270 | $570 |
| Climate | GPT-4o | **96.6%**[*] | — | 13.5M | $0 | $67,000 |
| | SIEVE (Ours) | **96.7%** | — | 7.5K | $180 | $220 |
| AI | GPT-4o | **95.5%**[*] | — | 13.5M | $0 | $67,000 |
| | SIEVE (Ours) | **95.7%** | — | 6K | $180 | $210 |
| Mainstream | GPT-4o | 92.4%[*] | 50% | 13.5M | $0 | $67,000 |
| | SIEVE (Ours) | 91.0% | **54%** | 100K | $400 | $900 |
| Quality | GPT-4o | 88.2%[*] | 50% | 13.5M | $0 | $67,000 |
| | SIEVE (Ours) | 86.3% | **53%** | 60K | $270 | $570 |

Table 1: Performance results of applying SIEVE on five highly specialized filters (see Appendix A). SIEVE can match or exceed GPT-4o's quality in terms of balanced accuracy and human preference. On the other hand, SIEVE saves more than 99% of the cost compared to using GPT-4o. See Section 5 for experiment details. [*]We assess GPT-4o's accuracy by measuring output consistency between identical API calls using greedy decoding (temperature set to 0). Some inconsistency persists, possibly due to hardware non-determinism. This may be amplified when using our CoT prompts.

The mainstream knowledge prompt aims to exclude any obscure and niche content determined by GPT-4o. Lastly, the quality filter aims to identify text snippets that are considered high quality by GPT-4o. Detailed prompts can be found in Appendix A.

Throughout our experiments, we use the encoder part of pretrained T5-large (Raffel et al., 2020) as the lightweight model. A linear layer is attached to the encoder model for binary classification. The classification model has less than 770M parameters, orders of magnitude smaller than GPT-4o. To mitigate the imbalanced nature of the snippets shown in Table 3, we utilize the focal loss (Lin, 2017). We refer the readers to Appendix C for more training details.

## 5.1 RESULTS: GPT-4O-BASED AND HUMAN-BASED EVALUATION

**GPT-4o-Based Evaluation.** Table 1 demonstrates that SIEVE achieves comparable balanced accuracy to GPT-4o on politics, climate, and AI filters, while closely matching its performance on mainstream knowledge and text quality filters. We evaluated these filters using a set of test snippets randomly sampled from OpenWebText. Ground truth labels were established through individual GPT-4o API calls for each snippet. To assess GPT-4o's performance against this ground truth, we conducted additional API calls using identical prompts for each test snippet, effectively measuring the model's self-consistency rate. Surprisingly, we observed inherent noise in GPT-4o's decisions, even when using greedy decoding (temperature set to 0). This variability likely stems from non-deterministic factors in the hardware infrastructure. While we employed chain-of-thought (CoT) reasoning in our filtering prompts (detailed in Appendix A) to enhance decision quality, the compounding noise from each generated token appears to have contributed to the noticeable inconsistencies. We also note that CoT plays an increasing role in inference-time scaling as demonstrated by OpenAI's o1 model, which may inevitably cause inconsistencies in the models' decisions.

**Human Evaluation.** In the above, we compared our distilled lightweight model's accuracy to GPT-4o for both mainstream and quality filters. When evaluated by GPT-4o itself, our model's performance appeared lower, raising the question: Is our lightweight model actually worse, or is GPT-4o biased when judging its own decisions? To investigate, we conducted a human evaluation. For each filter, we randomly selected 100 text snippets where GPT-4o and our lightweight model disagreed. We then recruited two groups of 13 annotators per filter to manually assess these challenging cases. The ground truth for each snippet was determined by the majority vote of the annotators, allowing us to compare both models' performance against human judgment. As shown in the fourth column of Table 1, we see even a slight edge of our lightweight model over GPT-4o when judged by human. This suggests our lightweight model is at least comparable to the decisions made by GPT-4o, while incurring much less computational cost when filtering web-scale datasets.

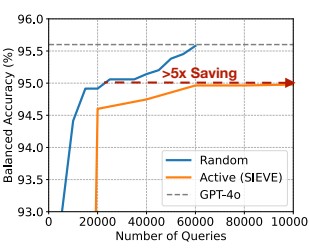 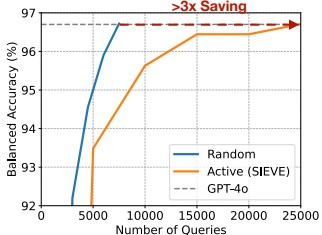 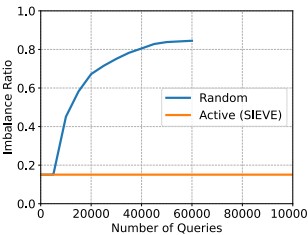

(a) Politics filter accuracy     (b) Climate filter accuracy     (c) Politics filter imbalance ratio of queried snippets

Figure 4: **Active vs Random Distillation**: Performance of the distilled lightweight model across different number of queries made to GPT-4o. With active learning, we can save the number of queries to GPT-4o by more than 5x for the politics filter and more than 3x for the climate filter.

| Filter | Method | Bal. Accuracy (GPT-4o as GT) | #Queries to GPT-4o | GPT-4o Cost | Lightweight Model Training | Lightweight Model Inference | Total Cost |
|---|---|---|---|---|---|---|---|
| Politics | GPT-4o | **95.6%**[*] | 13.5M | $67,000 | $0 | $0 | $67,000 |
| | SIEVE (Ours) | **95.6%** | 60K | $300 | $120 | $150 | $570 |
| | SIEVE (Ours) | **95%** | 25K | $125 | $30 | $150 | $305 |
| | Random | **95%** | 100K | $500 | $20 | $150 | $670 |
| Climate | GPT-4o | **96.6%**[*] | 13.5M | $67,000 | $0 | $0 | $67,000 |
| | SIEVE (Ours) | **96.7%** | 7.5K | $40 | $30 | $150 | $220 |
| | Random | **96.6%** | 25K | $125 | $20 | $150 | $315 |

Table 2: Cost breakdown of politics and climate filters. Total cost is consisted of GPT-4o querying cost, lightweight model training cost and T5 inference cost.

## 5.2 ACTIVE VS RANDOM DISTILLATION

In this section, we compare the effectiveness of active learning versus random querying for distilling our lightweight model. The random distillation strategy involves querying GPT-4 on a predetermined number of randomly selected snippets from the OpenWebText dataset, followed by fine-tuning the T5 encoder model on this queried data. As illustrated in Figure 4, our comparison of the lightweight model's performance when trained with active versus random sampling for both politics and climate filters reveals significant advantages for active distillation. Specifically, active distillation demonstrates remarkable efficiency, requiring over 5 times fewer queries for the politics filter and more than 3 times fewer for the climate filter compared to random distillation. Figure 4c also demonstrates our active learning algorithm's effectiveness in querying a much more balanced set of snippets. It's noteworthy that even with 100,000 queries, random distillation fails to match GPT-4o's performance. Due to budgetary constraints, we did not extend random distillation experiments beyond this query count. Importantly, we observe that as accuracy increases, active learning typically yields greater query budget savings compared to random sampling (Citovsky et al., 2021; Ash et al., 2021; Zhang et al., 2024a), suggesting that for the politics filter, in particular, the efficiency gains from active distillation could potentially exceed the observed 5-fold reduction in query costs. As inference cost of state-of-art LLMs increases, active distillation could play an increasingly important role in SIEVE.

## 5.3 COMPUTATIONAL COST COMPARISONS

**Cost Breakdown and Comparison.** Table 2 presents a comprehensive breakdown of SIEVE's computational costs, consisting of GPT-4o query costs, lightweight model training costs, and inference costs for filtering the entire OpenWebText dataset. The sum of GPT-4o query and training costs represents the total model distillation expense for SIEVE. Our experiments indicate that querying GPT-4o for 1000 snippets costs approximately $5. The lightweight model inference cost, a conservative estimate for processing 13.5M snippets using a 770M parameter model, is expected to be lower in practice by parallelization across multiple CPUs or cheap inference GPUs. The lightweight model training cost includes costs for model fine-tuning at each iteration of Algorithm 1, inference costs

for computing sigmoid scores of data in stream, and negligible uncertainty update expenses. We calculated these costs based on the hourly rate for 8xA100 80GB GPUs, multiplied by the actual time spent on each experiment. Together, these components provide a detailed overview of the computational resources required for SIEVE's implementation and application to the entire dataset.

As shown in Table 2, active distillation offers a more cost-effective approach compared to random distillation in achieving equivalent model performance. This cost advantage becomes even more pronounced when considering more expensive teacher models, such as the recently released o1, which significantly increases query costs and potentially dominates the total expenses. Given that active distillation substantially reduces the number of required queries, the cost disparity between active and random distillation methods is expected to widen further, especially when utilizing more advanced and costly teacher models.

**Stream-Based vs Pool-Based Active Distillation.** The choice of a stream-based active learning approach for SIEVE is justified by its significant cost advantages over pool-based methods. While stream-based algorithms process snippets only once, pool-based active learning requires repeated forward inference on the entire dataset of 13.5M snippets for each batch of queried snippets. This difference translates to substantial additional costs: approximately $1800 for the politics filter and $750 for the climate filter. These extra expenses would dramatically increase the overall cost of active distillation. Our decision to develop a stream-based active learning algorithm for SIEVE, particularly effective in imbalanced scenarios, is thus strongly supported by these cost considerations, ensuring a more economically viable solution for our system.

## 6 RELATED WORK

**Data Filtering for Large Language Models.** Data curation is fundamental to the development of LLMs (Longpre et al., 2023; Zhou et al., 2023). As interest in domain-specific LLMs grows, the need for extensive, relevant data collection becomes increasingly important. For a comprehensive overview of existing datasets, we direct readers to Raffel et al. (2020); Gao et al. (2020); Liu et al. (2024). Current methods for acquiring domain-specific data predominantly rely on a few established large-scale databases, including textbooks (Gunasekar et al., 2023), code repositories (Muennighoff et al., 2023; Gao et al., 2020), medical literature, and other specialized sources (Gao et al., 2020; Cheng et al., 2023). However, for rapidly evolving topics like the 2024 presidential election, climate change and artificial intelligence, relevant information is often dispersed across the internet, making comprehensive data collection challenging. Our approach involves training lightweight, task-specific data filtering models distilled from GPT-4. These models are then applied to web-scale datasets to identify pertinent information across various domains. Existing data filtering techniques span a wide range, from basic rule-based methods utilizing sentence-level statistical features (Rae et al., 2021; Yang, 2019; Laurençon et al., 2022; Zhang et al., 2022b) to sophisticated filters leveraging pretrained neural networks for text quality (Brown, 2020; Du et al., 2022; Chowdhery et al., 2023; Touvron et al., 2023; Enomoto et al., 2024; Qian et al., 2024) and toxicity (Lees et al., 2022; Friedl, 2023) filtering. To our knowledge, this work represents the first attempt to develop domain-specific data filtering models adaptable to a diverse array of filtering requirements and specialized domains.

**Active Learning** Active learning is a strategy aimed at reducing data annotation costs by selectively choosing which examples to label. Traditional approaches iteratively update machine learning models based on newly labeled data, using various informativeness metrics to guide the selection process. These metrics typically include uncertainty (Lewis & Gale, 1994; Tong & Koller, 2001; Settles, 2009; Balcan et al., 2006; Kremer et al., 2014; Gal et al., 2017; Ducoffe & Precioso, 2018; Beluch et al., 2018), diversity (Sener & Savarese, 2017; Geifman & El-Yaniv, 2017; Citovsky et al., 2021), and expected model change (Ash et al., 2019; 2021; Wang et al., 2021; Elenter et al., 2022; Mohamadi et al., 2022). However, most existing methods are designed for pool-based settings with balanced data distributions, which may not be suitable for all real-world scenarios.

In contrast, our work focuses on stream-based algorithms capable of handling data imbalance, an area that has received limited attention in deep learning contexts. While pool-based algorithms have shown success in addressing class imbalance (Aggarwal et al., 2020; Kothawade et al., 2021; Emam et al., 2021; Zhang et al., 2022a; Coleman et al., 2022; Jin et al., 2022; Cai, 2022; Nuggehalli et al., 2023; Zhang et al., 2024b; Lesci & Vlachos, 2024; Fairstein et al., 2024), stream-based approaches

for deep neural networks remain understudied. The recent work by Saran et al. (2023) introduces an online volume sampling technique for annotating diverse sets of examples in the representation space. Representation diversity, however, has been shown to struggle on class-imbalanced data distribution under the pool based setting (Zhang et al., 2022a; Nuggehalli et al., 2023; Lesci & Vlachos, 2024; Fairstein et al., 2024). Specifically these studies suggest that sampling diversely in the representation space does not necessarily improve the class-balancedness of the annotated examples. To address this gap, we propose what we believe to be the first stream-based algorithm specifically designed for class imbalance scenarios in active learning.

As part of our algorithm, we use an agnostic active learning algorithm for identifying the TRM threshold. Agnostic active learning has been widely studied in the classical PAC learning setups, where the labels of any particular example is inherently noisy. Our procedure is a direct application of Jamieson & Jain (2022), which was inspired by Dasgupta et al. (2007). The algorithm is proven to be near minimax optimal in these literature. In addition, our algorithm can also be seen as an instance of the algorithm proposed by Katz-Samuels et al. (2021) for threshold classifiers, where they also prove such algorithm is near instance-optimal. In this paper, we also prove the first bound towards balancedness of labeled examples in agnostic active learning, focusing on the class of threshold classifiers.

**Knowledge Distillation**   Knowledge distillation is a technique where a smaller "student" model learns to emulate a larger, more sophisticated "teacher" model. With the increasing capabilities of large language models (LLMs) across diverse tasks, recent research has explored using LLMs as annotators to train domain-specific student models. For a comprehensive review, see Tan et al. (2024). While most research in LLM knowledge distillation focuses on knowledge extraction methods, few studies address the high computational cost of using LLMs for large-scale annotation. Recent work by Zhang et al. (2023) and Rouzegar & Makrehchi (2024) has begun to tackle this issue by using active learning. Our paper applies knowledge distillation to the specific problem of data filtering. We employ a straightforward approach, using LLM annotations as binary classification labels to actively fine-tune an encoder-based model. Future research could explore using GPT-4's chain-of-thought outputs to distill a decoder-based student model within the multi-task learning framework proposed by Hsieh et al. (2023). It's worth noting that classic knowledge distillation work, such as Hinton (2015), trains student classifiers to match the teacher models' output logits. However, in our case, using chain-of-thought filtering prompts makes it impractical to obtain probabilities for the binary decision.

## 7   CONCLUSION, LIMITATIONS AND FUTURE WORK

In this paper, we introduced SIEVE, demonstrating the feasibility of achieving GPT-4o quality data filtering across a diverse range of user-specified filtering prompts. Our comprehensive study showcases the effectiveness of SIEVE in curating large-scale, high-quality datasets for language model training at a fraction of the cost of existing techniques. The experimental results, validated on the OpenWebText using five highly customized filter tasks, provide strong evidence of SIEVE's capability to match GPT-4o's accuracy while significantly reducing computational expenses.

While our initial study presents promising results, we acknowledge that there are several avenues for future enhancement and exploration. For future work, we are particular excited in scaling SIEVE to even larger datasets like the PILE, and more data modalities beyond text. The modular nature of SIEVE also allows for the integration of more advanced active learning algorithms. Additionally, SIEVE serves as an excellent testbed for these algorithms, offering immediate real-world impact. Future work could also investigate the incorporation of semi-supervised learning techniques to further reduce annotation costs following the framework proposed by Zhang et al. (2024a). Moreover, while our current implementation focuses on T5 architectures, future research could examine the efficacy of SIEVE with a broader range of pretrained model architectures for transfer learning. Lastly, exploring the use of more powerful models beyond GPT-4o, such as o1, for handling complex filtering prompts could extend the capabilities of SIEVE to even more challenging scenarios. In such cases, the importance of active learning becomes even more pronounced, as the increased querying costs associated with these advanced models necessitate highly efficient sampling strategies.

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

# A  FILTERING PROMPTS

## A.1  POLITICS

Please analyze the following text snippet and determine if it is relevant to aspects of a presidential election. The snippet may be arbitrarily cut off from a longer article, so please ignore any oddities caused by the truncation and focus on the overall relevance to presidential elections.

1. Read the text snippet carefully.
2. Identify any key terms, phrases, or concepts related to presidential election. These include by are not limited to Candidate information (biographies, backgrounds, policy positions)
   • Economic policies and their potential impacts
   • Social issues and proposed solutions
   • Foreign policy stances and international relations
   • Campaign events, debates, and public appearances
   • Polling data, electoral projections, and voter demographics
   • Media coverage, endorsements, and fact-checking
   • Campaign finance and fundraising efforts
   • Party dynamics and internal politics
   • Electoral processes, including voting systems and potential reforms
   • Controversies or scandals involving candidates or their campaigns
   • Vice presidential candidates and potential cabinet members
   • Analysis of key battleground states or regions
   • Digital campaigning strategies and social media presence
   • Grassroots organizing and volunteer efforts
   • External events or crises that may influence the election
3. Ignore any abrupt beginning or ending of the snippet and focus on the main content.
4. Assign either "PASS" for content relevant to presidential election and "FAIL" for those that are irrelevant.

Based on your analysis, determine if the snippet is relevant or irrelevant to the general knowledge of presidential elections. Think step by step and provide your final answer as either "PASS" or "FAIL" at the end of your response and nothing else.

Text snippet: <Insert Text Snippet>

## A.2  CLIMATE

Please analyze the following text snippet and determine if it is relevant to the general knowledge of climate change. The snippet may be arbitrarily cut off from a longer article, so please ignore any oddities caused by the truncation and focus on the overall relevance to climate change.

1. Read the text snippet carefully.
2. Identify any key terms, phrases, or concepts related to climate change, such as global warming, carbon emission, energy, technology innovation, agriculture, natural resources, pollution, landfill, chemistry, rising sea levels, extreme weather events, climate policies, environmental justice and sustainability.
3. Assess whether the snippet discusses causes, effects, or solutions to climate change, or provides information that contributes to the understanding of climate change.
4. Ignore any abrupt beginning or ending of the snippet and focus on the main content.
5. Assign either "PASS" for content relevant to climate change and "FAIL" for those that are irrelevant.

Based on your analysis, determine if the snippet is relevant or irrelevant to the general knowledge of climate change. Think step by step and provide your final answer as either "PASS" or "FAIL" at the end of your response and nothing else.

Text snippet: <Insert Text Snippet>

### A.3 AI

Please analyze the following text snippet and determine if it is relevant to aspects of AI. The snippet may be arbitrarily cut off from a longer article, so please ignore any oddities caused by the truncation and focus on the overall relevance to artificial intelligence.

1. Read the text snippet carefully.
2. Find content related to artificial intelligence that discusses computer systems or software designed to perform tasks typically requiring human intelligence, such as visual perception, speech recognition, decision-making, and language translation. Look for explanations of technologies enabling machines to learn from experience, adjust to new inputs, and perform human-like tasks without explicit programming. Include descriptions of systems that can interpret visual information from the world, as well as content about software capable of processing, analyzing, generating, or understanding human language. Seek information on machines or programs that can improve their performance through experience or data. Include discussions of AI applications in various fields such as healthcare, finance, transportation, education, or entertainment. Consider content addressing ethical considerations surrounding AI, including bias, privacy, job displacement, or long-term implications of advanced AI systems. Look for historical accounts of AI development, major milestones, breakthroughs, or setbacks in the field. Include explanations of AI algorithms, their workings, strengths, limitations, and potential applications. Capture debates or discussions about the future of AI, including topics like artificial general intelligence, superintelligence, or potential societal impacts of widespread AI adoption. Include reports on current research, new methodologies, experimental results, or theoretical advancements in AI. Consider content about prominent figures, organizations, or companies significantly contributing to AI research and development. Look for discussions of AI policy, regulation, or governance at organizational, national, or international levels. Include explanations of the relationship between AI and other fields such as robotics, Internet of Things, big data, or quantum computing. Finally, capture content addressing challenges in AI development, such as data quality, computational requirements, or the need for explainable AI systems.
3. Ignore any abrupt beginning or ending of the snippet and focus on the main content.
4. Assign either "PASS" for content relevant to AI and "FAIL" for those that are irrelevant.

Based on your analysis, determine if the snippet is relevant or irrelevant to aspects of artificial intelligence, its development, applications, or implications. Think step by step and provide your final answer as either "PASS" or "FAIL" at the end of your response and nothing else.

Text snippet: <Insert Text Snippet>

### A.4 MAINSTREAM KNOWLEDGE

**We asked for obscure knowledge instead, so any snippet that "Failed" would be considered mainstream knowledge.**

Please analyze the following text snippet and determine if it contains obscure or niche knowledge that less than 10000 people know and understand. The snippet may be arbitrarily cut off from a longer article, so please ignore any oddities caused by the truncation and focus on the overall relevance to artificial intelligence.

1. Read the text snippet carefully.
2. Filter the dataset for content related to obscure, specialized, or highly niche knowledge that is not commonly known or easily accessible to the general public. Include information on rare historical events, obscure scientific theories, uncommon philosophical concepts, extinct languages, highly specialized mathematics, niche literary works, rare medical conditions, uncommon species, esoteric subcultures, mystical practices, experimental technologies, obscure laws, rare geological formations, lesser-known art movements, specialized crafting techniques, rare musical instruments, uncommon culinary practices, obscure sports, theoretical cosmological concepts, and highly specialized areas of archaeology or anthropology. Exclude any content that is commonly known, part of standard education, regularly discussed in popular media, or widely understood by the general public. The ideal content should require specialized knowledge, extensive research, or access to uncommon sources of information, rather than being something an average person would encounter in daily life or through casual exposure to media and education.

3. Ignore any abrupt beginning or ending of the snippet and focus on the main content.
4. Assign either "PASS" for obscure and niche knowledge and "FAIL" for those that are common knowledge.

Think step by step. Then, you must provide your final answer as either "PASS" or "FAIL" at the end of your response and nothing else.

Text snippet: <Insert Text Snippet>

A.5 QUALITY

Please analyze the following text snippet and determine if it is high quality or low quality training data for a large language model. The snippet may be cut off abruptly from a longer piece of text, but focus your analysis on the quality factors present in the provided text rather than the awkward truncation. Quality factors to consider include:

1. Evaluate the spelling, grammar, and overall writing quality of the snippet. Note any errors or inconsistencies that could negatively impact the model's learning.
2. Assess the factual accuracy and reliability of the information presented in the snippet. Consider whether the content appears trustworthy and well-researched.
3. Analyze the clarity, coherence, and logical flow of ideas in the snippet. Determine if the text is easy to understand and follow.
4. Gauge the breadth and depth of knowledge conveyed in the snippet. Consider whether the content provides valuable information or insights on the topic at hand.
5. Examine the neutrality and objectivity of the tone and perspective presented in the snippet. Consider if the text appears biased or presents a balanced viewpoint.
6. Based on the above factors, determine if the snippet is:
   PASS: High quality training data
   FAIL: Low quality training data

Think step by step and answer with either PASS or FAIL as your final decision in the end and nothing else.

Text snippet: <Insert Text Snippet>

# B  ANALYSIS PROOF

*Proof.* Let $A = \sum_{j:f(x_j)\in[\underline{\mu},\bar{\mu}]} \eta_j$ and $B = \sum_{j:f(x_j)\in[\underline{\mu},\bar{\mu}]} 1 - 2\eta_j$, we can rewrite the imbalance ratio into $\lambda(\underline{\mu},\bar{\mu}) = \frac{A}{A+B}$. Since $N(\gamma_1 - \gamma_0) = \sum_{j:f(x_j)\in(\underline{\mu},\bar{\mu}]} 1 - 2\eta_j \geq B - LN$, we can lower bound the balancedness by $\frac{A}{A+N(\gamma_1-\gamma_0)+LN}$.

As the lower bound $\frac{A}{A+N(\gamma_1-\gamma_0)+LN}$ increases as $A$ increases, we would now like to prove a lower bound of $A = \sum_{j:f(x_j)\in[\underline{\mu},\bar{\mu}]} \eta_j$.

Recall $\pi(1), ..., \pi(N)$ is the ordering of examples based on sigmoid score. We let $\pi^{-1}(\cdot)$ denote the inverse mapping of $\pi$, so that $\pi^{-1}(\pi(i)) = i$. We let $\underline{r} = \min(\{j : f(x_{\pi(j)}) \in [\underline{\mu}, s^\star]\})$, $\bar{r} = \max\{j : f(x_{\pi(j)}) \in (s^\star, \bar{\mu}]\}$ and $r^*$ denote the index where $f(x_{\pi(r^\star)}) = s^\star$.

First note since class 0 is the minority class, we must have $r^\star \neq N$. Since $r^\star \neq 0$ and $r^\star \neq N$, we must have $\eta_{\pi(r^\star)} \geq 0.5$ and $\eta_{\pi(r^\star+1)} \leq 0.5$. Otherwise, $r^\star + 1$ or $r^\star - 1$ will have lower risk than $R(\eta_{\pi(r^\star)})$. By the smoothness definition above, we further have $\forall j \leq r^\star, 0.5 - (r^\star - j)L \leq \eta_{\pi(j)} \leq 0.5 + (r^\star - j + 1)L$, and $\forall j \geq r^\star, \eta_{\pi(j)} \geq 0.5 - (j - r^\star + 1)L$.

$A = \sum_{j:f(x_j)\in[\underline{\mu},\bar{\mu}]} \eta_j$ can then be rewritten in the ranked format as $A = \sum_{j\in[\underline{r},\bar{r}]} \eta_j$.

First, we divide the sampling range into $[\underline{r}, r^\star]$ and $[r^\star + 1, \bar{r}]$. When sampling in $[\underline{r}, r^\star]$, we have

$$R(f(x_{\pi(\underline{r})})) - R(s^\star) = \gamma_0 > 0 \implies$$
$$N\gamma_0 = \sum_{j\in(\underline{r},r^\star]} \eta_{(j)} - \sum_{j\in(\underline{r},r^\star]} (1 - \eta_{(j)}) \geq -1 + \sum_{j\in[\underline{r},r^\star]} \eta_{(j)} - \sum_{j\in[\underline{r},r^\star]} (1 - \eta_{(j)}). \quad (3)$$

When sampling in $[r^\star + 1, \bar{r}]$, since $R(\bar{r}) - R(r^\star) = \gamma_1$, we must have

$$N\gamma_1 = N \cdot (R(\bar{r}) - R(r^\star)) = \sum_{j\in(r^\star,\bar{r}]} 1 - 2\eta_{(j)} = \sum_{j\in[r^\star+1,\bar{r}]} (1 - \eta_{(j)}) - \sum_{j\in[r^\star+1,\bar{r}]} \eta_{(j)}. \quad (4)$$

To obtain the lower bound of $\sum_{j\in[\underline{r},\bar{r}]} \eta_{(j)}$, we start by bounding $\bar{r}$ and $\underline{r}$. Specifically, by equation 4, we have

$$N\gamma_1 = \sum_{j\in[r^\star+1,\bar{r}]} (1 - 2\eta_{(j)}) \leq \sum_{j\in[r^\star+1,\bar{r}]} (1 - 2(0.5 - (j - r^\star)L))$$
$$= \sum_{j\in[r^\star+1,\bar{r}]} 2(j - r^\star)L = \sum_{j=1}^{\bar{r}-r^\star} jL = (\bar{r} - r^\star)(\bar{r} - r^\star + 1)L \leq L(\bar{r} - r^\star + 1)^2.$$

As a result $\bar{r} \geq \sqrt{\frac{N\gamma_1}{L}} + r^\star - 1$. Let $\alpha_1 = \sqrt{\frac{N\gamma_1}{L}} - 1$, we then have $\bar{r} \geq r^\star + \alpha_1$.

Similarly, we have

$$N\gamma_0 \leq \sum_{j\in[\underline{r},r^\star]} (2\eta_{(j)} - 1) \leq \sum_{j\in[\underline{r},r^\star]} (2 \cdot (0.5 + (r^\star - j + 1)L) - 1)$$
$$= \sum_{j\in[\underline{r},r^\star]} 2(r^\star - j + 1)L = \sum_{j=1}^{r^\star-\underline{r}+1} 2jL \leq (r^\star - \underline{r} + 2)^2 L,$$

so $\underline{r} \leq r^\star - \sqrt{\frac{N\gamma_0}{L}} + 2$. Let $\alpha_0 := \sqrt{\frac{N\gamma_0}{L}} - 2$, we then have $\underline{r} \leq r^\star - \alpha_0$

Now, we can bound $\sum_{j \in [r^\star + 1, \bar{r}]} \eta_{(j)}$ by the following

$$\sum_{j \in [r^\star + 1, \bar{r}]} \eta_{(j)} = \sum_{j = r^\star + 1}^{r^\star + \alpha_1} \eta_{(j)} \geq \sum_{j = r^\star + 1}^{r^\star + \alpha_1} 0.5 - (j - r^\star)L$$

$$= \sum_{j=1}^{r^\star + \alpha_1} 0.5 - jL = \frac{\alpha_1}{2} - \frac{\alpha_1(\alpha_1 + 1)L}{2}$$

$$= \frac{(1 - L)\alpha_1 - \alpha_1^2 L}{2}.$$

Similarly, we can bound $\sum_{j \in [\underline{r}, r^\star]} \eta_{(j)}$ by the following

$$\sum_{j \in [\underline{r}, r^\star]} \eta_{(j)} = \sum_{j = r^\star - \alpha_0}^{r^\star} \eta_{(j)} \geq \sum_{j = r^\star - \alpha_0}^{r^\star} 0.5 - (r^\star - j)L$$

$$= \sum_{j=0}^{\alpha_0} 0.5 - jL = \frac{\alpha_0 + 1}{2} - \frac{\alpha_0(\alpha_0 + 1)L}{2}$$

$$= \frac{(1 - L)\alpha_0 - \alpha_0^2 L + 1}{2}$$

Together, we have

$$\sum_{j \in [\underline{r}, \bar{r}]} \eta_{(j)} \geq \frac{(1 - L)(\alpha_0 + \alpha_1) - (\alpha_0^2 + \alpha_1^2)L + 1}{2}.$$

$$\geq \frac{1}{2}((1 - L)(\alpha_0 + \alpha_1) + 1) \geq \frac{1}{2}(2(1 - L)\sqrt{\frac{N\underline{\gamma}}{L}} - 2) = (1 - L)\sqrt{\frac{N\underline{\gamma}}{L}} - 1$$

.

For the edge case, since the interval must have more than three snippets, we can bound $\sum_{j \in [\underline{r}, \bar{r}]} \eta_{(j)} \geq 1.5 - 2L$. Therefore, we have $\sum_{j \in [\underline{r}, \bar{r}]} \eta_{(j)} \geq \max(1.5 - 2L, (1 - L)\sqrt{\frac{N\underline{\gamma}}{L}} - 1)$.

Finally, we can bound the balancedness by

$$\frac{\sum_{j \in [\underline{r}, \bar{r}]} \eta_{(j)}}{\sum_{j \in [\underline{r}, \bar{r}]} (1 - \eta_{(j)})} \geq \frac{\sum_{j \in [\underline{r}, \bar{r}]} \eta_{(j)}}{\sum_{j \in [\underline{r}, \bar{r}]} \eta_{(j)} + N(\gamma_1 - \gamma_0) + LN}$$

$$\geq 1 - \frac{N(\gamma_1 - \gamma_0) + LN}{\sum_{j \in [\underline{r}, \bar{r}]} \eta_{(j)} + N(\gamma_1 - \gamma_0) + LN}$$

$$\geq 1 - \frac{N\bar{\gamma} + LN}{\sum_{j \in [\underline{r}, \bar{r}]} \eta_{(j)} + N\bar{\gamma} + LN}$$

$$\geq 1 - \min(\frac{N\bar{\gamma} + LN}{1.5 + N\bar{\gamma} + LN - 2L}, \frac{N\bar{\gamma} + LN}{(1 - L)\sqrt{\frac{N\underline{\gamma}}{L}} + N\bar{\gamma} + LN - 1})$$

$$\geq 1 - \min(\frac{N\bar{\gamma} + LN}{1.5 - 2L}, \sqrt{L} \cdot \frac{N\bar{\gamma} + LN + 1}{(1 - L)\sqrt{N\underline{\gamma}}})$$

$\square$

## C TRAINING DETAILS

Our model is fine-tuned using the AdamW optimizer with a cosine learning rate schedule. For every training of $f$ in Algorithm 1, we train up to 5 epochs. When measuring performance, we use a separate validation set to find the highest performance checkpoint. For focal loss, we use $\gamma = 5$, and $\alpha$ is set to the imbalance ratio estimated from Table 3 for the minority class.

