# OpenReview forum: "SIEVE: General Purpose Data Filtering System Matching GPT-4o Accuracy at 1% the Cost"
_ICLR.cc/2025/Conference — ICLR 2025 Conference Withdrawn Submission_

### Official Review · Reviewer_Bhab · 2024-11-02

**Soundness:** 3
**Presentation:** 3
**Contribution:** 3
**Rating:** 5
**Confidence:** 3

**Summary:**

The SIEVE system is designed to make large-scale data filtering more cost-effective by leveraging GPT-4o’s high-quality filtering capabilities as a guide for fine-tuning a more efficient, lower-cost T5 model. SIEVE's goal is to minimize reliance on GPT-4o by enabling T5 to approximate GPT-4o's filtering decisions, thus reducing computational costs.

**Strengths:**

+The paper addresses an important and practical challenge in the community: domain-specific data filtering.

+It introduces a TRM threshold approximation method to ensure balanced and informative data selection, effectively mitigating potential bias induced by class imbalance in datasets.

+The proposed approach is comprehensively validated across five datasets, comparing the performance and cost-effectiveness of GPT-4o and a lightweight T5 model.

+Theoretical support is provided to demonstrate the balance of labeled snippets, strengthening the method’s foundation.

More specifically:

SIEVE uses an active learning approach to select the most informative samples for GPT-4o annotation. Rather than using all available data, SIEVE prioritizes examples based on T5’s uncertainty in predictions. This uncertainty is measured with a True Risk Minimizer (TRM) threshold, which targets samples near the model’s decision boundary (i.e., where T5’s confidence is low). Selecting samples around this threshold ensures balanced, informative data that challenges the model, improving generalization without overwhelming it with redundant or obvious examples.

Recognizing that datasets can be imbalanced. Specifically, in filtering tasks, “fail” samples often outnumber “pass” samples. To address this issue, SIEVE implements a class-balanced sampling strategy. This approach uses the TRM threshold to select samples from both classes near the decision boundary, and ensures that training data has sufficient representation from both “pass” and “fail” classes, preventing T5 from becoming biased towards majority-class predictions.

**Weaknesses:**

-Fine-tuning is conducted incrementally with new batches of informative data labeled by GPT-4o, rather than retraining on a combined dataset that includes original T5 training data. In other words, SIEVE does not use the original T5 dataset during fine-tuning; instead, it focuses exclusively on samples curated through GPT-4o labeling. Will this approach cause potential catastrophic forgetting?

-The paper has several errors or unclear statements that could be misleading. For instance, in the last paragraph of Section 2.1, the authors mention training the T5-decoder, but in other sections, the T5-encoder is referenced. Additionally, in Figure 4, the legends for 'Random' and 'Active (SIEVE)' appear to be swapped. The meaning of the two 'SIEVE (Ours)' results for the Politics Filter in Table 2 is also ambiguous.

-There may be a disconnect between the goal of approximating the TRM threshold (and the interval of the confidence set) and the actual training methodology. While the approximation aims to balance the probabilities of majority and minority classes within the interval, the real training data may still be imbalanced due to sample distribution. If focal loss is employed to address this imbalance, it raises questions about the necessity of estimating the confidence set interval. Furthermore, the method of determining the alpha parameter in focal loss remains unclear. Using the imbalance ratio from randomly sampled snippets as alpha may not be appropriate or feasible in practical applications.

**Questions:**

1. What impact would training the model using a fixed sigmoid threshold of 0.5 have on the performance of the lightweight model? Could it degrade the model's effectiveness?
2. How does the upper and lower bound of the confidence set interval change throughout the training process? Could it lead to a situation where data samples selected as informative at the start of training fall outside the updated confidence interval, resulting in the model including less informative samples over time?
3. How is the initial labeled set in Algorithm 1 obtained? Is it initialized as an empty set, or is there a specific strategy for selecting the initial samples?
4. Do you only use (snippet, annotation) pairs to train the model or include the prompt as part of the input?

---

> ### Author Response · Authors · 2024-11-15
>
> Thank you for your insightful review and pointing out the unclear points of our paper. We address your concerns below.
>
> **Catastrophic forgetting:** we would like to note that we are doing a standard transfer learning process here, finetuning a pretrained T5 encoder model for a specific task of data filtering. As this finetuned model is only used for data filtering and not text generation, it is actually beneficial for the model to only keep the relevant parameters for the filtering task. As an example, in a domain specific case, if we are training a filtering model for medical knowledge, the model doesn't need to remember information about graduate level math.
>
> **Unclear statements:** thank you very much for pointing these out. It should be T5 encoder model throughout. And yes, the plot has the two curves switched. We will update the legends. The two rows with 'SIEVE (Ours)' both use the exact same process and only differ by the total GPT-4o query budget spent as indicated by the fourth row. The first 'SIEVE (Ours)' with a 60K budget is chosen so its accuracy matches GPT-4o, while the second 'SIEVE (Ours)' with a 25K budget is chosen so its accuracy matches the random baseline at 100K.
>
> **About imbalance ratio**, we are using the randomly chosen GPT4o calls at the initialization phase (line 183) to estimate the imbalance ratio. As these GPT4o queries need to be made already, we do not incur any additional cost and is certainly feasible in practice. Note that we are only using an estimate of the imbalance ratio instead of the true imbalance ratio, and this estimate can be very accuracy with only a few hundred random samples (by tail bounds of binomial distribution).
>
> In addition, the use of focal loss actually places the decision boundary more biased towards the majority class, which results in more unbalanced sampling during active learning. This is because, focal loss and the imbalanced classification literature in general, aims to shift the decision boundary towards where a balanced testing distribution of the two classes equalize in density. We refer the reviewer to classics like [1] for their work. In our case, we would like to shift the boundary to TRM threshold, where imbalanced unlabeled distributions of the two classes roughly equalize in density. In this case, we would like to shift the decision boundary more towards the minority class. As a concrete evidence, in [this figure (https://ibb.co/z7wDK9K)](https://ibb.co/z7wDK9K), we plot the ratio of unlabeled minority class examples at different sigmoid scores trained with focal loss (for the final model checkpoint of the quality filter result). A ratio of .5 means the minority and majority class examples equalize. It is clear that the decision threshold of .5 given by the focal loss is extremely biased towards the majority class. On the other hand, the TRM threshold would be where the y-axis reaches around .5, which is very different from the decision boundary given by the focal loss.
>
> So to summarize, when labeling around .5, we will definitely label a much less balanced dataset, which hinders the model performance based on prior literature in deep active learning under pool based settings. We are also launching an ablation study that select data closest to .5, and will include the discussions in the final version of our paper.
>
> **Upper and lower bounds**: these bounds are updated on line 197. This is a direct application of the [empirical Bernstein bound](https://arxiv.org/abs/0907.3740), which ensures the TRM threshold is always within the bounds across different epochs with at least 95% probability.
>
> In addition, the point of doing active learning is to train a new model on the latest labeled examples that are uncertain for the current model. Once the we update the current model with training on these uncertain examples, they will become well learned (certain) for the newly updated model. Therefore, the goal is exactly to labeled examples that are uncertain and balanced (within the upper and lower bounds), so that once we train on them, they become easy and certain examples for our latest model. In such cases, these examples will definitely fall out the lower and upper bounds, which is exactly what we want. The ideal ultimate goal is for there to be no snippets that fall into these lower and upper bounds (which entails 100% model accuracy).
>
> **Initial labeled set**: as suggested by our notation on line 183, these snippets are sampled uniformly at random. We will add some text to make this clearer.
>
> **Finetuning pairs**: due to the long prompts that we use, they do not fit into the context window of T5, so we are only training on the (snippet, annotation) pairs.
>
> **Thanks again for your review**, and we would love to hear your thoughts given our response.
>
> [1] Cao, K., Wei, C., Gaidon, A., Arechiga, N., & Ma, T. (2019). Learning imbalanced datasets with label-distribution-aware margin loss. Advances in neural information processing systems, 32.

---

### Official Review · Reviewer_jtzu · 2024-11-03

**Soundness:** 3
**Presentation:** 1
**Contribution:** 2
**Rating:** 5
**Confidence:** 3

**Summary:**

This paper proposes SIEVE for filtering web-scale data for the creation of clean datasets.
SIEVE utilized GPT-4o and proposed an active learning and knowledge distillation pipeline to distillate knowledge to lightweight T-5 Model for more cost-efficient filtering. SIEVE achieved comparable performance as GPT-4o on filtering OpenWebText dataset and with approximately 1% of the cost.

**Strengths:**

- The paper provides formal proofs and theoretical guarantees for their active learning algorithm's performance, particularly regarding class balancedness.
- The approach to filtering data for dataset creation is crucial for many LLM and DL tasks.
-  Achieves comparable performance to GPT-4o at with much less cost

**Weaknesses:**

- The system's performance is bounded by GPT-4o's capabilities and inherent inconsistencies
- The evaluation is a little bit limited and only conducted on OpenWebText
- The filtering is distillate from another LLM, so it's hard to guarantee the filtering robustness and fairness.
- While the paper demonstrates effectiveness on OpenWebText, but sounds like lacks novelty and it's combining of active learning with knowledge distillation.

**Questions:**

- The 1% cost compared to GPT-4o, my understanding is that you considering only inference cost, how about the whole training and active learning cost of the T-5 model?
- The human evaluation was conducted only on cases where GPT-4o and the lightweight model disagreed. But there still have possibility that both models make mistakes. Wouldn't it be more comprehensive to also evaluate a random sample of cases where they agreed?
- Why choose the T-5 model, and how was this architectural choice made? Shouldn't decoder-only architecture be more efficient in dealing with such filtering tasks, which has much higher throughputs?
- Given that your approach relies on GPT-4o's chain-of-thought reasoning, I am concerning about the reliability and of the LLM output. Probably authors may considering multi-agent (multi prompts) ensemble as the base teacher model.

---

> ### Author Response · Authors · 2024-11-14
>
> Thank you for your insightful review and pointing out the clarity issues in our paper. We respond to your concerns below:
>
> 1. Is active learning cost considered? Yes, in Table 1, lightweight model cost is the total cost of model training from active learning and inference cost on the entire dataset. In Table 2, this is specifically broken down into two separate costs of training and inference. We realize Table 1 is unclear, and we will add this detail in the paper. Thank you for pointing this out.
>
> 2. We agree that collecting human evaluation on snippets the two models agree on would enhance the results. The core message we wanted to demo here is that the lightweight model is able to achieve the same accuracy as GPT-4o. Since the accuracy is the same on snippets they agree on, we opted to spend our limited annotation budget on annotating the disagreement cases. If accepted, we will conduct additional human studies on the agreement cases as suggested by the reviewer.
>
> 3. Encoder architecture is still the most prevalent model for classification. Decoder models are more efficient from a memory standpoint (large decoder models can fit into GPUs but encoder models cannot). However, in our case, since the lightweight models are so small, they can fit into a GPU regardless of decoder or encoder architecture. In terms of total FLOPs during inference, decoder models only saves marginally over encoder models. Therefore, we decided to use an encoder based model as it attends to every token equally while only cost less than 10% of computation (in terms of FLOPs).
>
> 4. We agree that using stronger models like the multi-agent approach or even o1 can benefit the filtering quality. In such cases, the cost of GPT-4o queries will be even larger. Given the same query budget, it is an interesting question whether learning from a larger amount of noisy single GPT-4o queries would underperform a smaller amount of multi-agent GPT-4o queries. From our experience, more annotations from noisy single queries will be a more effective use of the query budget. But we agree with the reviewer this is something interesting to figure out, so we are starting an ablation study to compare the two scenarios that we will include in the final draft of the paper. Regardless the results, our system is not limited to any single teacher model (GPT4o was the state-of-art model at the time of writing), so it could definitely be adapted for more advanced teacher models as well.
>
> 5. Filtering robustness and fairness. Could we kindly ask the reviewer to elaborate what aspects of robustness and fairness you are referring to? From our perspective, robustness and fairness of any language model is impossible to guarantee. It is not clear to us which aspects of these metrics can be guaranteed for GPT-4o but not our lightweight model. We would be happy to include this in the limitation section, but are currently unclear on what specific aspects the reviewer is referring to.
>
> 6. In terms of novelty, we believe data filtering is a crucial research area as noted by the reviewer. Traditionally, data filters are trained on small quality/toxicity datasets annotated by human, leaving a large void of possibilities for domain specific and criterion specific data filters. From the perspective of data filtering methods, we believe our proposed methodology is very novel. In addition, from a technical side, the stream-based active learning algorithm that deals with imbalance and our theoretical bounds are all novel contributions to the field.
>
> Thanks again for your review, and we would love to hear your thoughts given our response.

---

### Official Review · Reviewer_DNz2 · 2024-11-04

**Soundness:** 4
**Presentation:** 3
**Contribution:** 3
**Rating:** 5
**Confidence:** 3

**Summary:**

The authors present SIEVE, a cost-efficient data filtering approach that uses stream-based active learning and reduces the number of filtering calls to costly models( in this case, gpt4) in favor of a lighter, finetuned model (T5). This approach reduces the cost to 1% while maintaining a similar accuracy to gpt4 on the OpenWebText dataset using five filter tasks.

**Strengths:**

This paper studies the high cost of using large models such as gpt4. This is a fascinating topic and should be studied more. That being said, these are some of the strengths of this paper:
1. Theoretical Analysis: This paper's strength is its addition of a theoretical section that provides a rigorous analysis of the balancedness bound for the TRM threshold.
2. Human Evaluation: The paper doesn't only consider gpt4 as the golden truth but also uses human evaluation to validate the results, which  strengthens the author's point.
3. Clear layout of the approach: The active learning approach is clearly presented and easy to understand.

**Weaknesses:**

Here are some places for improvement:
1. A detailed discussion on how $B$  from algorithm A was chosen and how it affects the performance.
2. The impact of model size on performance is not studied. Given that only T5 was used for experiments, I wonder how choosing a different model architecture or size would affect the performance both in terms of accuracy and cost (in terms of gpt 4 calls).
3. Are the legends for Figure 4 flipped? It looks like Active learning requires more queries.
4. How are you selecting the upper and lower bounds? Are there examples of values for those upper bounds for a specific filter?

**Questions:**

1. Why did you choose T5 specifically?
2. Have you considered any tasks beyond filtering that could benefit from your approach? Generalizing this to involve more tasks would be highly beneficial.

---

> ### Author Response · Authors · 2024-11-15
>
> Thank you for your insightful review and pointing out the unclear parts of our paper. We would like to address your concerns below.
>
> 1. The choice of $B$: Thank you for point this out. This is indeed an important parameter that deserves more documentation. The choice of $B$ is highly application and scenario dependent. Specifically, a larger $B$ means we need to retrain the model fewer number of times, thus introducing much lower training cost for the lightweight model finetuning. At the same time, a larger $B$ with less number of model updates also enjoys less of the accuracy benefit from active learning. In practice, we believe $B$ should be chosen based on the lightweight model training budget. If the lightweight model is trained on machines one owns, it costs close to nothing every time the model is retrained. In such cases, $B$ should be chosen to be smaller, so that query cost of GPT-4o can be reduced. We also think in practice the practitioners should choose $B$ adaptively, based on the validation accuracy the model obtains thus far. The practitioner would be recommended to start with a smaller $B$ (i.e. 1.5K), and switch to larger $B$ (i.e. 5K or even 20K) if the model is still far away from a target level of accuracy. The validation set can simply consists of 1K to 5K of random annotated samples.
>
> In our experiments, we set $B$ to be $1.5K$ initially and increased it to $5K$ if the model accuracy did not achieve the target accuracy after 10 batches.
>
> 2. We fully agree with the reviewer that studying different architectures can strengthen our paper. We are launching an experiment with distillbert and will include the results and discussion in the final version of our paper. We will also try to report the results before the discussion period ends. We chose $T5$ for no apparent reasoning other than it is one of the latest encoder-based open source model.
>
> 3. Yes, the legends are flipped. We will fix that.
>
> 4. The upper and lower bounds are updated on line 197, and is a direct application of the [empirical Bernstein bound](https://arxiv.org/abs/0907.3740). This simple ensures the TRM threshold is always within the bounds across different epochs with at least 95% probability. The bounds are dynamically and automatically adjusted for each filter and each iteration of the active learning process.
>
> 5. Applying SIEVE beyond data filtering is definitely an important future work and we will update the future work section. We would like to note that data filtering itself is a crucial problem in training LLMs today, also noted by all other reviewers. Therefore, we believe the scope of our work is addressing a sufficiently important problem already. Extending this approach to more broad application is definitely going to make a large impact, but falls out of the scope of this paper.
>
> Thanks again for your review, and we would love to hear your thoughts given our response.

---

### Official Review · Reviewer_pcM9 · 2024-11-04

**Soundness:** 3
**Presentation:** 2
**Contribution:** 2
**Rating:** 5
**Confidence:** 3

**Summary:**

The paper introduces SIEVE, a cost-effective system for high-quality data filtering that achieves results comparable to GPT-4o while operating at just 1% of its cost. SIEVE employs a T5-based binary classification model to replicate GPT-4o’s filtering decisions, classifying text snippets as relevant (pass) or irrelevant (fail). To minimize expenses, SIEVE integrates an active learning approach that selectively queries only the most informative snippets for labeling by GPT-4o. This targeted approach reduces the need to label every snippet, gathering only the most valuable samples. Each batch of GPT-4o-labeled data is used to periodically fine-tune the T5 model, progressively improving its accuracy in replicating GPT-4o’s filtering.

To address class imbalance—where the majority of snippets may not meet filtering criteria—the active learning algorithm incorporates a True Risk Minimizer (TRM) threshold. Unlike traditional uncertainty sampling, which uses a fixed 0.5 threshold and can over-represent the majority class, the TRM threshold dynamically adjusts to balance samples from both majority and minority classes around the decision boundary. This design ensures a more balanced labeled dataset, reducing the risk of model bias. The TRM threshold is approximated using a high-confidence interval, updated iteratively as new snippets enter in a streaming format, allowing the model to query only the most promising samples.

This efficient and adaptive approach enables SIEVE to create a balanced and accurate training set for the T5 model with minimal reliance on costly GPT-4o queries. The result is a scalable, high-accuracy filtering system that optimizes computational resources and cost, making SIEVE well-suited for large-scale data filtering applications in training language models.

**Strengths:**

1. The methodology is sound. Incremental fine-tuning and the TRM threshold enhance efficiency, though testing on more diverse datasets would better demonstrate generalizability.

2. SIEVE’s cost-efficient filtering method is impactful for large-scale data filtering, achieving GPT-4o-level quality at minimal cost.

**Weaknesses:**

**1. Limited exploration of diverse datasets.**

An area where SIEVE could improve is in testing across a broader range of datasets. While the OpenWebText dataset provides a good baseline for evaluating filtering performance, testing SIEVE on datasets with varied writing styles, content types, or noise levels would offer a clearer picture of its versatility. For instance, it would be helpful to see how SIEVE performs on datasets with different imbalance ratios or topics to understand whether the TRM threshold effectively adapts to various content structures and domain-specific challenges.

**2. Baseline limitations.**

Although SIEVE is compared to GPT-4o, adding baselines with more straightforward methods (e.g., T5-based filtering or basic binary classifiers without active learning) would give a clearer insight into SIEVE's improvements. This would help show how much each part—like the active learning framework and TRM threshold—contributes to its performance. If a basic T5 model alone performs comparably, it could suggest that the added complexity of SIEVE may not be necessary. Including T5 as a baseline would help clarify whether SIEVE’s specific design choices are essential for its performance improvements or if similar results could be achieved with a more straightforward approach.

**3. Unclear writing.**

The paper does not provide details on how the sigmoid score is calculated for determining snippet relevance. Since the TRM threshold relies on the sigmoid score to identify the informative samples, it’s necessary to provide a clear explanation of how the model and approach are used for calculating this score.

**Questions:**

1. Would it be possible to include comparisons with simpler filtering methods, like T5-based classifiers?

2. Has SIEVE been tested on domain-specific datasets (e.g., medical or scientific)? If not, could you discuss any anticipated adjustments needed to maintain performance in these areas?

3. Could you clarify the model or method used to calculate the sigmoid score necessary for determining the TRM threshold?

---

> ### Author Response · Authors · 2024-11-15
>
> Thank you for your insightful review and pointing out the unclear parts of our paper. We would like to address your concerns below.
>
> 1. Dataset settings: Even though we only tested on the OpenWebText dataset, different filters we tested on have induced classification tasks of different imbalance ratio (see Figure 3, which ranges from 0.026 to 0.457). In addition, as indicated by the self-consistency accuracy in Table 1, GPT-4o has noise levels anywhere from 3% to 12% across different filtering tasks. We would also like to note that the OpenWebText is a web crawl of the data on the internet, which already contains diverse topics and types of data online, including books, news articles, online forums, and more. We choose the OpenWebText dataset because of its broad coverage and its reasonable size for academic research. To our knowledge, it is hard to find a web crawl dataset of this particular scale (~10B tokens). In addition, this is the dataset GPT2 was trained, giving this dataset a lot of credibility.
>
> 2. T5 baselines: We totally agree with the reviewer we should include this as a baseline. We are running the experiment and do not expect T5 to perform well at all on this task (this can be seen by the poor performance when T5 is only finetuned on a few queries from GPT4o in Figure 4).
>
> 3. Thank you for pointing out the clarity issue when calculating the sigmoid scores. Described in Algorithm 1, our active learning algorithm iteratively trains a new neural network model based on the $B$ newly labeled samples at each outer iteration. This model $f$ is trained and denoted on line 185. The sigmoid scores are simply applying the latest lightweight model $f$ on the unlabeled snippets coming into the stream, i.e., $f(x)$ for any new snippet x. This is also detailed on line 194. We will make this more clear in our text.
>
> 4. When filtering domain specific datasets, as this is likely a subset of the web crawl already, we expect the approach taken by SIEVE to work as well. We do think leveraging domain specific foundation models could save additional queries to GPT4o than T5. We will add this discussion to the future work section.
>
> Thanks again for your review, and we would love to hear your thoughts given our response.

---

### Note · Authors · 2024-11-22

I have read and agree with the venue's withdrawal policy on behalf of myself and my co-authors.